# Effect of Material Inhomogeneity on the Crack Tip Mechanical Field and SCC Growth Rate of 52M/316L Dissimilar Metal Welded Joints

Kuan Zhao *, Bangwen Wang *, He Xue and Zheng Wang

School of Mechanical Engineering, Xi'an University of Science and Technology, No. 58 Yanta Middle Road, Xi'an 710054, China
* Correspondence: xinkuan1022@xust.edu.cn (K.Z.); wangbangwen2206@126.com (B.W.)

**Abstract:** The stress–strain conditions at the crack tip in dissimilar metal welded joints (DMWJ) are a critical factor influencing stress corrosion cracking (SCC) behavior. The processing technology and working environment of DMWJ lead to a randomly inhomogeneous distribution of material mechanical properties, making the crack tip mechanical field more complex. An inhomogeneous model was obtained using a combination of physical experiments and the elastic–plastic finite element method to understand the effect of this inhomogeneous distribution of mechanical properties on the direction of SCC growth and the growth rate in DMWJ and the impact of inhomogeneity on the SCC growth behavior was compared and analyzed. The findings demonstrate that Type I (opening mode) cracks are more likely to form due to the inhomogeneity of mechanical properties and are more likely to deflect toward the Alloy 52M region at the interface between Alloy 52M and 316L stainless steel. Additionally, the strain gradient at the crack tip increases with the degree of inhomogeneity, which has a bigger impact on the accuracy of SCC growth rate predictions.

**Keywords:** dissimilar metal welded joints; material inhomogeneity; mechanical properties; elastic–plastic finite element





## 1. Introduction

Over the last three decades, many attempts have been made to understand the underlying mechanism of environmentally assisted cracking (EAC) in the components and structures of pressurized water reactors (PWR) because it is critical to explore the remaining life of primary structures in nuclear power plants [1,2]. Dissimilar metal welded joints (DMWJ) are frequently utilized in nuclear power PWR primary circuit systems to link the reactor pressure vessel to the primary circuit mains. Due to the structural characteristics and harsh service environment of DMWJ, it has a high sensitivity to environmental damage. It is susceptible to EAC represented by stress corrosion cracking (SCC), which will significantly impact the safe operation and service life of nuclear power equipment [3–6]. However, the inhomogeneity of the material's inherent organization and the welding process leading to a high degree of organizational and mechanical properties inhomogeneity in the local and interfacial regions are among the main factors affecting the behavior of SCC [7–9]. Therefore, it is critical to investigate the crack tip mechanical field under the inhomogeneity of material mechanical properties to quantitatively predict the direction of SCC crack growth behaviors and growth rate (da/dt), which has been the focus of scholars and engineers.

Researchers have proposed various mechanisms and prediction models to study SCC crack extension [10–12]. Many scholars have recognized the slip/dissolved oxidation model as a reasonable SCC description for important structures in nuclear power plants [13]. The strain rate at the crack tip is an essential parameter in this model for expressing a mechanical condition, although it is more difficult to acquire. Xue et al. [14,15] proposed a method to

quantitatively predict the crack extension rate of SCC by replacing the crack tip strain rate with the normal plastic strain rate at the characteristic distance. However, the accuracy of the prediction results is directly related to the selection of the appropriate feature distance. Because of the inherent properties of DWMJ, the mechanical properties of various zones of the material become complex and heterogeneous, causing mechanical properties such as strength, hardening index, and elastic modulus to vary somewhat in the same region and have a significant impact on the structural integrity of critical PWR components [16–18]. Zhu et al. analyzed the microstructure of 52M-316L welded joints. The results showed that the Vickers hardness, grain size, grain boundary characteristics, and residual strain in the region near the weld fusion line differed, exhibiting some inhomogeneity [19].

In addition, by analyzing the effect of yield strength mismatch at the weld fusion line boundary (FB) on the strength of welded joints, some scholars [2,20] found that weld strength mismatch can have an important effect on weld SCC. And the SCC phenomenon is more likely to occur in the FB compared to other regions. However, an efficient way to guarantee the safe functioning of nuclear power welded joints is by accurately measuring the mechanical properties of key components. The mechanical properties of different regions of DMWJ have been obtained by indentation tests, microhardness, or micro-sample tensile tests [3,4,21]. Ming et al. [21] found that the mechanical properties of welded joints in different locations have a large variation, and the strength of the material shows a similar trend to the hardness. The Weibull distribution has been frequently utilized to characterize the heterogeneity of welded joint materials to explore the impact of this random inhomogeneous distribution of mechanical characteristics on the SCC of welded joints [22,23]. However, little information is available on how these random inhomogeneous distributions of mechanical properties affect the mechanical state of the DMWJ crack tip.

Since the factors associated with the inhomogeneity of the DMWJ material are random, the mechanical properties of the material also vary with the internal spatial location of the material. This work used the Vickers hardness test to determine the local mechanical properties of the DMWJ and then combined the elastic-plastic fracture mechanics method with the finite element method to explore the impact of inhomogeneity on the crack tip mechanical field and SCC growth rate.

## 2. Methods and Calculation Model

### 2.1. Weibull Distribution of Mechanical Properties of Welded Joint Materials

Some unstable factors influence DMWJ materials in the manufacturing and actual engineering processes, which makes the materials have certain non-uniformity. In turn, the mechanical properties of the material, such as hardness, yield strength, and strain hardening index, vary with the spatial location within the material, and the mechanical properties of the material itself are random due to the non-uniformity of the distribution of the material affected by the welding process [24].

It has been shown that the Weibull distribution is applicable in describing the inhomogeneous characteristics of the internal mechanical properties of the material [22,23]. Therefore, the mechanical properties of the microscopic units composing the material in this study satisfy the Weibull distribution again, and the probability density function of the Weibull distribution with two parameters is used as follows:

$$f(x) = \frac{\lambda}{x_0} \left( \frac{x}{x_0} \right)^{\lambda-1} \exp \left( -\frac{x}{x_0} \right)^{\lambda} \tag{1}$$

where $x$ is the random variable of the distribution (yield strength, modulus of elasticity, and other mechanical properties), $x_0$ is the scale parameter; $\lambda$ is the shape parameter that characterizes the material's degree of inhomogeneity.

The parameters $x_0$, $\lambda$ of the Weibull distribution in Equation (1) satisfy the following relationships with the mean $\mu$ and variance $\sigma^2$ of the distribution:

$$\begin{cases} \mu = x_0 \Gamma\left(1 + \frac{1}{\lambda}\right) \\ \sigma^2 = x_0{}^2\left[\Gamma\left(1 + \frac{2}{\lambda}\right) - \Gamma^2\left(1 + \frac{1}{\lambda}\right)\right]^{\lambda > 0,\ x_0 > 0} \end{cases} \tag{2}$$

It is easy to see from Equation (2) that as $\lambda$ increases, the more the value of the $\Gamma$ function tends to 1, the closer $x_0$ is to the mean value $\mu$, the lower is the non-uniformity of the material.

### 2.2. Specimen and Material Model

Compact tensile specimens with a constant load $K_I$, as depicted in Figure 1, are extensively employed for SCC experiments in high-temperature water conditions [25]. Therefore, 1T-CT specimens with prefabricated cracks were used for numerical tests in this study to investigate the impact of randomness in the mechanical properties of materials on the SCC growth rate. The numerical test procedure was performed under the guidance of the American Society for Testing and Materials Standards [26].

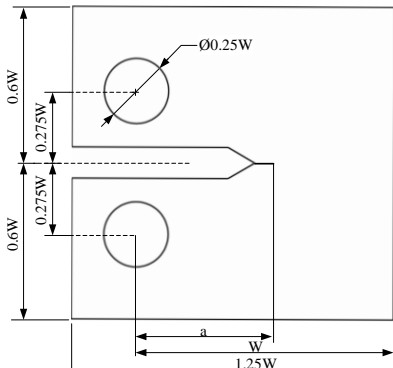

**Figure 1.** 1T-CT specimen geometry (where W = 50 mm, a = 0.5 W, B is specimen thickness B = 0.5 W).

To study the influence of random material mechanical properties on the fracture performance of welded joints, the SA508-52M-316L type DMWJ, which is widely used in the safety end of the AP1000 type III nuclear primary circuit, was selected as the test object in this study.

The primary constituent materials are 316L stainless steel for the base material at the pipe end, nickel-based Alloy 52M for the weld metal and SA508 alloy steel for the base material at the pressure vessel end. According to the sampling location of the welded joint, three crack locations were selected in this study; crack 1 was located in the nickel-based Alloy 52M, crack 2 was located in the interface area between the nickel-based Alloy 52M and 316L stainless steel, and crack 3 was located in the 316L stainless steel. The prefabricated crack length was 2 mm, as shown in Figure 2.

The stress–strain relationship of the material in DWMJ is expressed as the Ramberg–Osgood equation [20], and the specific expression can be written as:

$$\frac{\varepsilon}{\varepsilon_y} = \frac{\sigma}{\sigma_y} + \alpha\left(\frac{\sigma}{\sigma_y}\right)^n \tag{3}$$

In Equation (3), $\sigma_y$ is the yield strength, $\varepsilon$ is the true strain, $\sigma$ is the true stress, $\varepsilon_y$ is the yield strain, $\alpha$ is the offset coefficient, $n$ is the strain hardening exponent of the material, and the mechanical properties of the material are shown in Table 1 [16,17]. Generally, the material inhomogeneity in DMWJ has little influence on the elastic modulus and Poisson's ratio, which are assumed constant in physical experiments and simulation studies [27]. Vickers hardness tests were used in this work to get the Vickers hardness HV of various

locations along the fusion line of nickel-based Alloy 52M with 316L stainless steel in DMWJ to explore the randomness of the mechanical properties of the material more precisely. The load used in the experiments for the Vickers hardness test was 0.9807 N (100 gf). In addition, to ensure the accuracy of the hardness test, the experiment was guided by the international Vickers hardness test standard [28]. The test was conducted at room temperature, and 5 columns were set up on each side of the fusion line, with 10 test points in each column, each column was spaced 1 mm apart, and each test point was set at an interval of 2 mm, as shown in Figure 3. The Vickers hardness data was measured as shown in Figure 4.

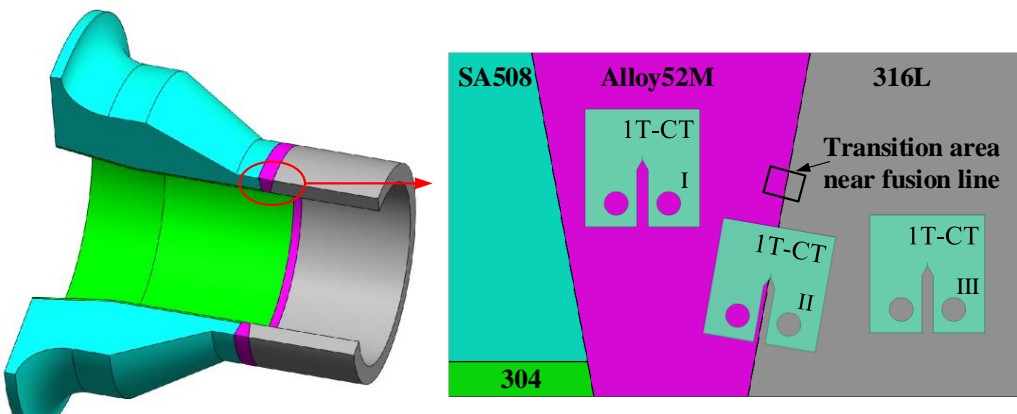

**Figure 2.** Welded joint sampling and crack location diagram (where I—52M specimen: crack 1, where II—weld line specimen: crack 2, where III—316Lspecimen: crack 3).

**Table 1.** Mechanical property of the material [16,17].

| Material | Young's Modulus, *E* (MPa) | Poisson's Ratio, *ν* | Yield Offset, *α* | Hardening Exponent, *n* |
|---|---|---|---|---|
| 52M | 180,000 | 0.3 | 9.33 | 4.99 |
| 316L | 176,390 | 0.3 | 10.46 | 3.24 |

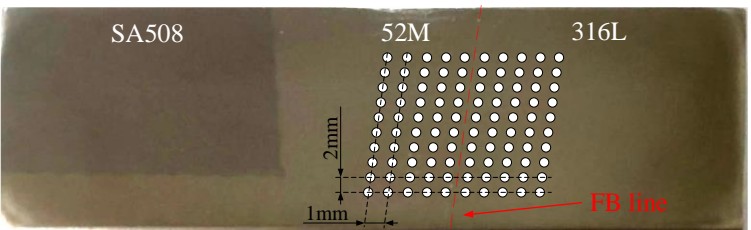

**Figure 3.** Vickers hardness test samples and hardness test points.

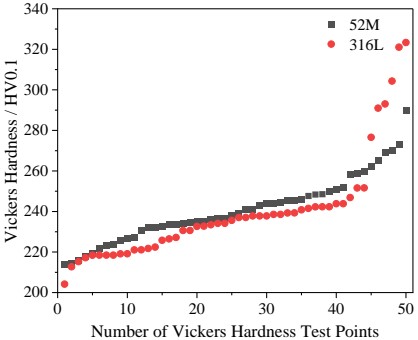

**Figure 4.** Vickers hardness data point diagram.

Peng et al. [29] described the linear relationship between Vickers hardness HV0.1 and yield strength for the HAZ of DMWJ using the following equation:

$$\sigma_y = 3.28\text{HV} - 211 \tag{4}$$

The yield strength of the weld metal nickel-based Alloy 52M can be obtained from the following equation [30]:

$$\sigma_y = 3.15\text{HV} - 168 \tag{5}$$

The yield strength of 50 test points each in 316L stainless steel and nickel-based Alloy 52M were calculated by Equations (4) and (5). Assuming that the yield strength of these two materials obeys the Weibull distribution, the Weibull statistical analysis is then performed on the test point samples. The fitted Weibull distribution density function curves are shown in Figure 5. The scale parameter $x_{01}$ = 571.22 and shape parameter $\lambda_1$ = 11.01 for 316L stainless steel. scale parameter $x_{02}$ = 615.12 and shape parameter $\lambda_2$ = 11.89 for Alloy 52M.

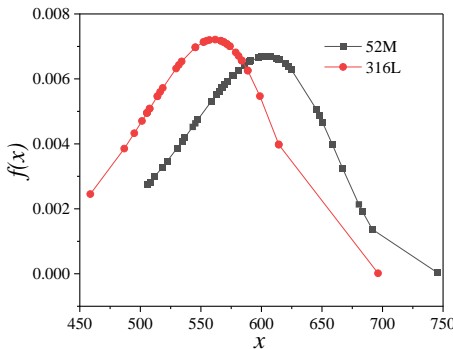

**Figure 5.** Weibull statistical distribution density function curve of yield strength.

With the results of the above analysis, this paper implements the definition of non-homogeneity of welded joint materials on the ABAQUS platform (ABAQUS V6.14, 2015, Dassault Systèmes, Vélizy-Villacoublay, France) through two stages: first, random samples of yield strength satisfying the Weibull distribution are generated for each of the two materials. Second, the random sample data are mapped into the finite element model. The mapping approach taken in this paper is a one-to-one assignment of data to cells, which means that each cell corresponds to random data and is isotropic and homogeneous within each cell. These two steps achieve an inhomogeneous material conforming to the Weibull distribution.

In summary, this paper implements the Weibull distribution of random inhomogeneous material finite element model to meet the following steps.

(1) By the Vickers hardness conversion to get the yield strength of the welded joints at multiple points;
(2) Fit the data to obtain the parameters of the Weibull distribution;
(3) Extract the unit information of the finite element model;
(4) For each unit to generate a random sample of a given Weibull distribution;
(5) Write the random sample data into the bcell material information.

### 2.3. Finite Element Model

Numerical simulations were conducted with the commercial finite element code ABAQUS, expecting to represent the stress and strain at the crack tip throughout the SCC experiments. In the SCC experiments, the crack front in the thickness direction of the specimen is mostly driven by plane strain conditions. Hence, the specimen is simplified to a plane strain model. The sub-model technique was applied in this calculation to investigate and acquire the local stresses and strains at the fracture tip. The global model and sub-model containing prefabricated cracked specimens are shown in Figure 6a,b, respectively, where

the X-axis is the direction of crack growth and the Y-axis is the direction normal to the crack growth. The global model and sub-model mesh numbers are 31,140 and 11,863, respectively, using eight-node quadrilateral quadratic plane strain cells. In the numerical simulation process, the stress intensity factor $K_I$ is usually taken as the mechanical parameter. A constant load is taken as the loading condition, $K_I$ set to a constant 30 MPa m$^{1/2}$ [31].

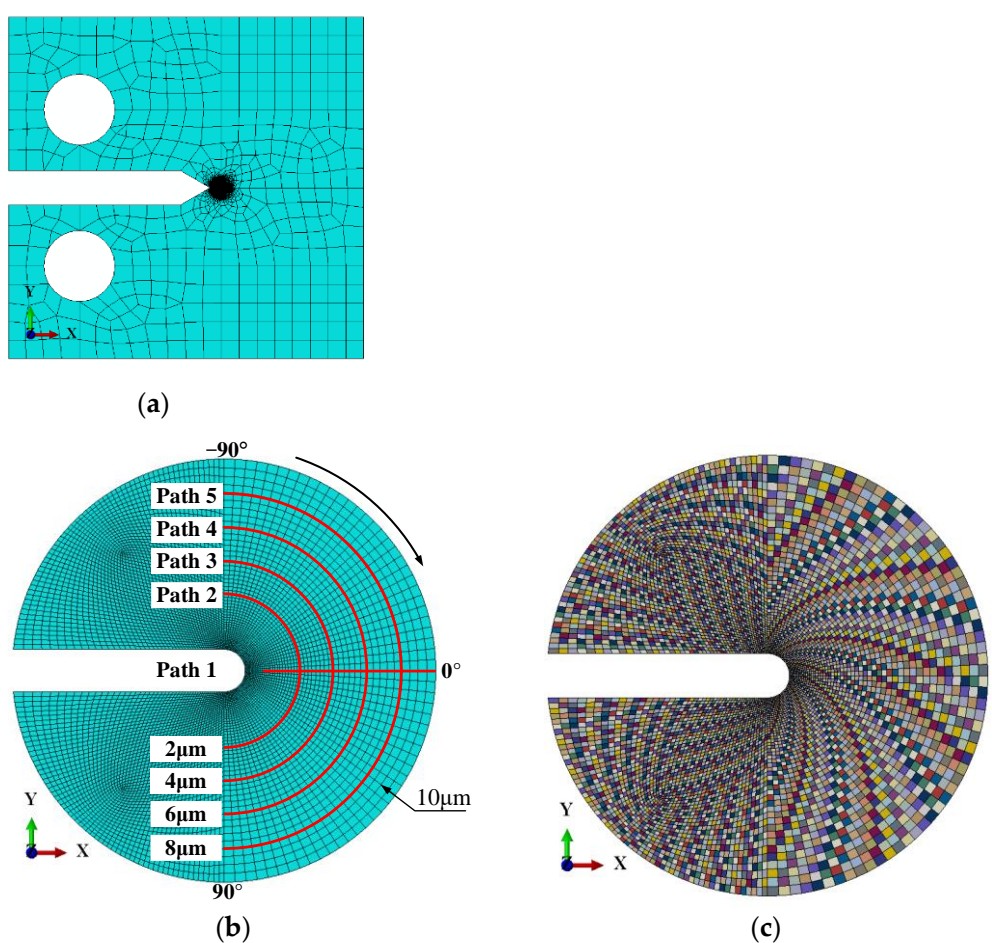

**Figure 6.** Finite element model of the CT specimen. (**a**) Global model, (**b**) sub-model and observation path. (**c**) Details around the crack tip of the heterogeneous material.

In order to provide a detailed characterization of the effect of random inhomogeneous material on the stress and strain around the crack tip, the crack growth direction and circumferential direction can be used as characteristic paths [32]. The normal plastic strain at the crack tip is a key parameter in predicting the crack extension rate, and the crack jump distance ranges from 1 to 10 μm [33]. Thus, path 1 was obtained as a path from 1 μm to 10 μm from the crack tip along the crack extension direction. Paths 2, 3, 4, and 5 were extracted from −90° to 90° at 2, 4, 6, and 8 μm from the crack tip, respectively, according to Figure 7a. The inhomogeneity of the DMWJ material is achieved by assigning different materials to each cell, according to Figure 7b, where various gray levels represent various material properties.

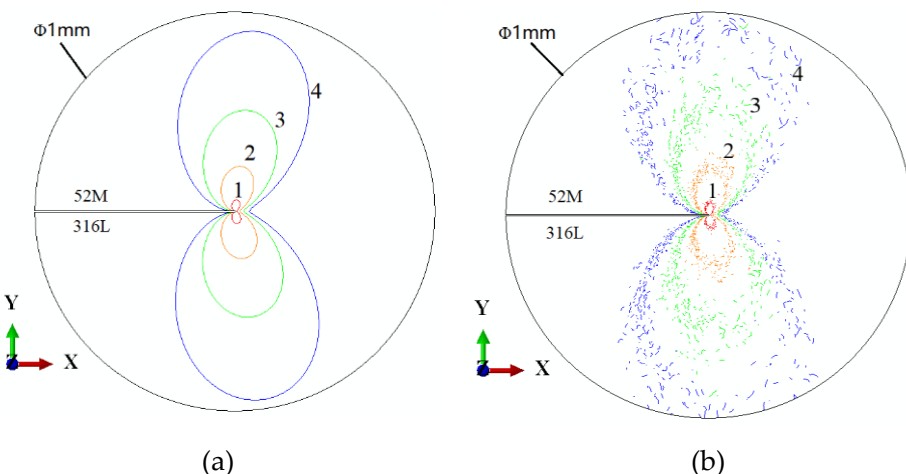

**Figure 7.** Plastic zone at the crack tip of the fusion line specimen (where equivalent plastic strain is 0.2% and contours 1, 2, 3, and 4 are the plastic zone boundaries for applied loads $K_I$: (1) $K_I$ = 10, (2) $K_I$ = 20, (3) $K_I$ = 30, (4) $K_I$ = 40 MPa m$^{1/2}$). (**a**) Homogeneous material; (**b**) inhomogeneous material.).

## 3. Results and Discussion

### 3.1. Plastic Zone at the Crack Tip of the Fusion Line Specimen

Local mechanical factors in SCC experiments will significantly impact the quantitative prediction of SCC growth rates. Therefore, this paper will investigate the effect of randomly distributed inhomogeneous materials on mechanical parameters such as local plastic strain and plastic zone at the crack tip. A circular region with a crack tip diameter of 1 mm was analyzed. Normally the equivalent strain of 0.2% is taken as yield strain; Figure 7 shows the plastic zone near the fused boundary line specimens for homogeneous and inhomogeneous materials, where contours 1, 2, 3, and 4 are the plastic zone boundaries for applied loads $K_I$ of 10, 20, 30, and 40 MPa m$^{1/2}$, respectively. Comparing Figure 7a,b, it is obvious that homogeneous and inhomogeneous materials have the same trend of the plastic zone under the same loading conditions. Still, the inhomogeneous model has a larger range of plastic zone at the crack tip and a more dispersed boundary under the same loading conditions. As the applied load increases, the dispersion of the boundary of the plastic zone of the non-homogeneous material will also be a more significant gradient of variation. Since plastic strain acts as the main parameter affecting the crack extension rate, the effect of non-homogeneity of the material is even more not negligible.

### 3.2. The Strain Field Feature around the Crack 1 Tip

Figure 8a,b show the equivalent plastic strain within 1 mm of the tip of crack 1 in both homogeneous and inhomogeneous Alloy 52M. It is evident that the equivalent plastic strain zone around the crack tip is larger in the inhomogeneous material compared to the homogeneous material at the same contour. The plastic zone boundary is more dispersed. The equivalent plastic strain in homogeneous materials is symmetrically distributed about the crack tip, while inhomogeneous materials exhibit a certain degree of asymmetry. Figure 9a,b show the normal plastic strains at the crack tips in the homogeneous and inhomogeneous Alloy 52M. Similar to the distribution of the equivalent plastic strain in Figure 8, the distribution of the normal plastic strain at the tip of crack 1 in the inhomogeneous material also exhibits asymmetry and dispersion.

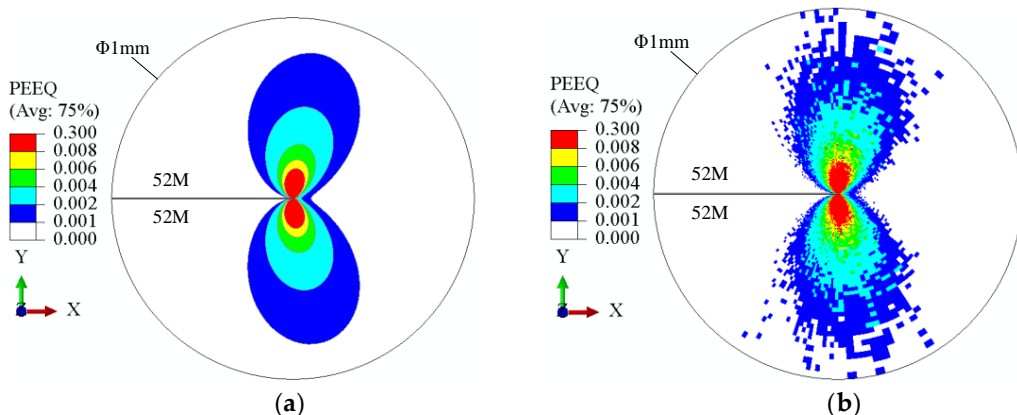

**Figure 8.** Equivalent plastic strain at the tip of crack 1: (**a**) homogeneous; (**b**) inhomogeneous.

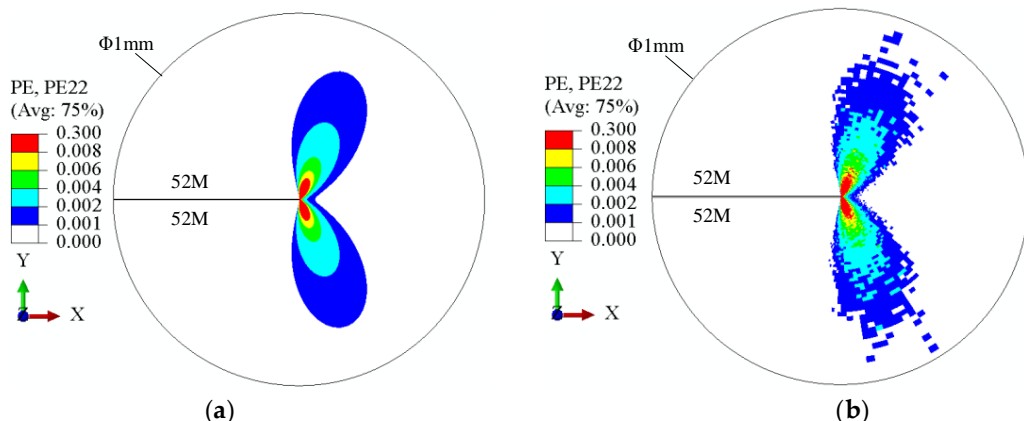

**Figure 9.** Normal plastic strain at the tip of crack 1: (**a**) homogeneous; (**b**) inhomogeneous.

To more clearly represent the effect of this inhomogeneity on the plastic strain, Figure 10 shows the distribution of the normal plastic strain for multiple observed paths in the two material models, where *m* represents the degree of inhomogeneity of the material, and the larger λ, the closer the material is to homogeneity. Figure 10a depicts the −90° to 90° normal plastic strain around the crack tip when the distance r from the crack tip is 2, 4, 6, or 8 μm. Comparing the normal strain curves of homogeneous and inhomogeneous materials at 2 μm from the crack tip, it is clear that the homogeneous material is symmetrical at approximately 0° and has a maximum value at 45°. In contrast, the inhomogeneous material is not symmetric at 0°, and the maximum value is not at 45°. As the distance *r* increases, the normal plastic strain at the crack tip becomes smaller, and the closer the curve is to the homogeneous curve, the smaller the gradient of variation will be. The normal plastic strain of crack 1 along path 1 is shown in Figure 10b, where the inhomogeneous has a larger normal plastic strain at the crack tip compared to the homogeneous. These results suggest that the SCC crack growth rate (CGR) may increase in inhomogeneous materials, and that increasing non-homogeneity in yield strength has a greater impact on the accuracy of predicting the CGR in Alloy 52M.

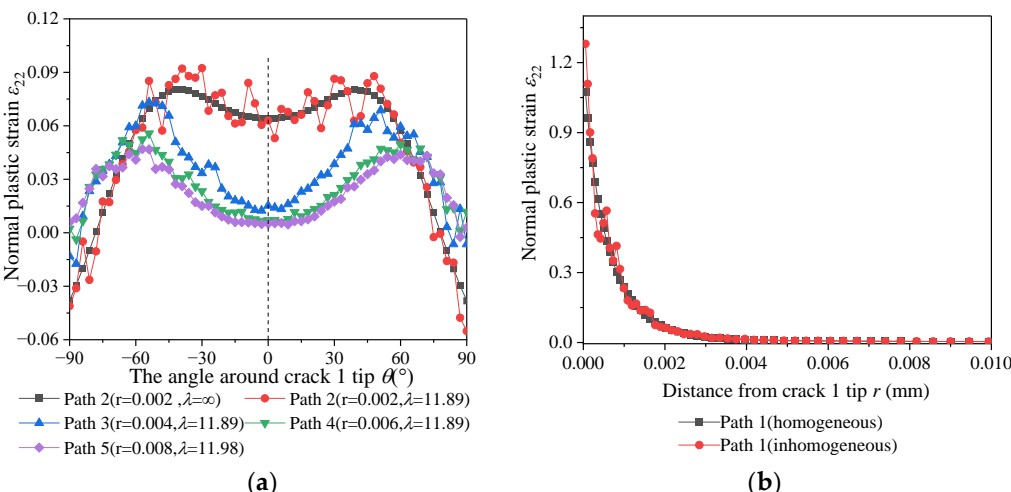

**Figure 10.** Normal plastic strain under homogeneous and inhomogeneous conditions: (**a**) around the crack 1 tip; (**b**) in front of the crack 1 tip.

### 3.3. The Strain Field Feature around the Crack 3 Tip

Figure 11a,b show the equivalent effect variation clouds in the 1 mm range of the crack tip in homogeneous and inhomogeneous 316L stainless steel. Similar to the equivalent plasticity of the crack tip in Alloy 52M, the range of equivalent plastic strain zones in the same contour of the crack tip is larger in inhomogeneous materials than in homogeneous materials and is asymmetrically distributed in the X-axis direction. Figure 12a,b show the normal plastic strains of cracks in homogeneous and inhomogeneous 316L stainless steel specimens. Comparing Figures 9b and 12b, it is obvious that the ones in the 316L alloy specimens are smaller than those in the Alloy 52M specimens, indicating that the CGR in the Alloy 52M may be faster than that in the 316L alloy under the same load conditions.

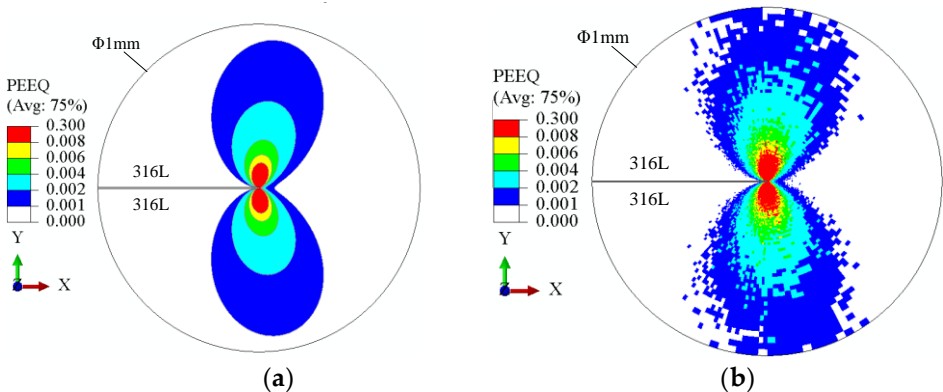

**Figure 11.** Equivalent plastic strain at the tip of crack 3: (**a**) homogeneous, (**b**) inhomogeneous.

Figure 13 displays the distribution of normal plastic strain for multiple observed paths in homogeneous and inhomogeneous 316L specimens. Figure 13a shows the normal plastic strain at different distances from the circumferential direction of the crack tip from $-90°$ to $90°$. Comparing the path 2 curves in homogeneous and inhomogeneous specimens, the homogeneous path 2 curves are symmetric at about $0°$ and show a minimum value in the $0°$ direction and a maximum value in the $45°$ direction. The normal plastic strain curve of path 2 in the inhomogeneous specimen is asymmetrically distributed, and the distribution of both maximum and minimum values is different from that of the homogeneous model. The gradient of the normal plastic strain fluctuation of the crack will become smaller and smaller as the radius distance of the path circumference increases. Comparing path 2 in the inhomogeneous model in Figures 10a and 13a, It is evident that the fluctuation range of the normal plastic strain curve along path 2 in 316L stainless steel is larger than that in

Alloy 52M due to the different inhomogeneity coefficients λ. And the higher is the degree of inhomogeneity, the greater is the fluctuation of the normal plastic strain, and the greater the impact on the accuracy of predicting the crack extension rate. The normal plastic strains along path 1 for both specimens are shown in Figure 13b. The normal plastic strain gradient of the crack in the inhomogeneous model is larger than that in the homogeneous model, which indicates that the SCC crack extension rate in the inhomogeneous 316L stainless steel may be faster than that in the homogeneous material.

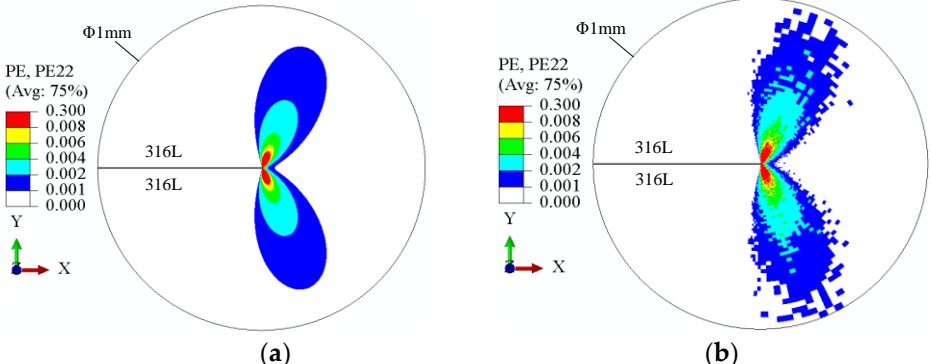

**Figure 12.** Normal plastic strain at the tip of crack 3 (**a**) homogeneous (**b**) inhomogeneous.

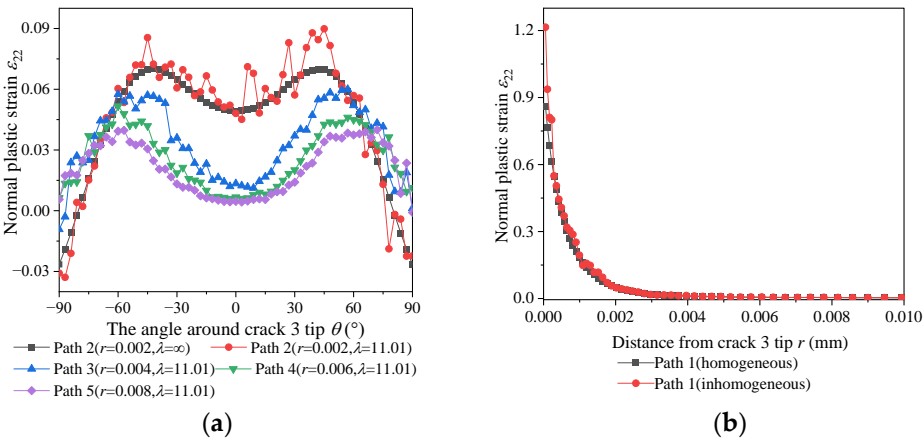

**Figure 13.** Normal plastic strain under homogeneous and inhomogeneous conditions: (**a**) around the crack 3 tip; (**b**) in front of the crack 3 tip.

*3.4. The Strain Field Feature around the Crack 2 Tip*

Figure 14a,b depict the 1 mm equivalent plastic strain clouds around the crack tip in the homogeneous and inhomogeneous fusion boundary specimens, respectively. The equivalent plastic strains on either side of the fusion line crack tip of the homogeneous and inhomogeneous specimens were asymmetrically distributed. The strains in alloy 52M were more significant than those in the 316L stainless steel. The strain zone of the inhomogeneous fusion line specimen under the same contour is larger than that of the homogeneous specimen. Figure 15a,b show the normal plastic strains in the homogeneous and inhomogeneous fusion line specimens, which, similar to the equivalent plastic strains, show an asymmetric distribution at the interface of the two materials. These results indicate that the strain is not homogeneous in the inhomogeneous fusion boundary line specimens, which may lead to the variation of the crack extension direction on the fusion line, and the extension direction is easily transferred to the Alloy 52M.

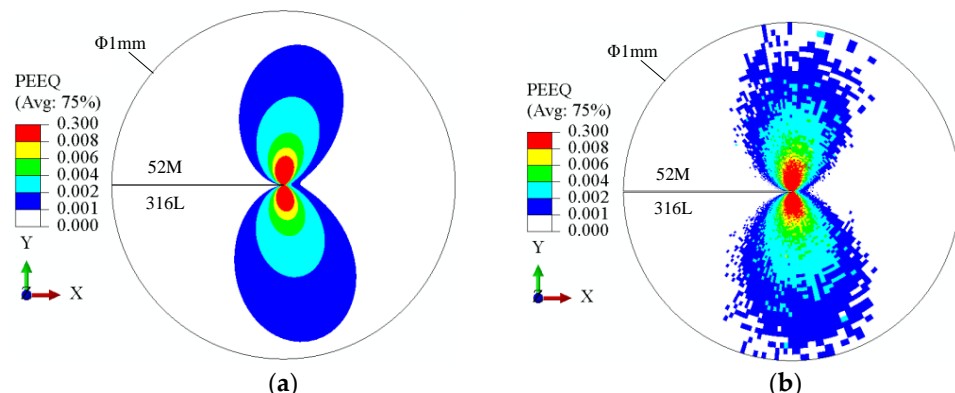

**Figure 14.** Equivalent plastic strain at the tip of crack 2: (**a**) homogeneous; (**b**) inhomogeneous.

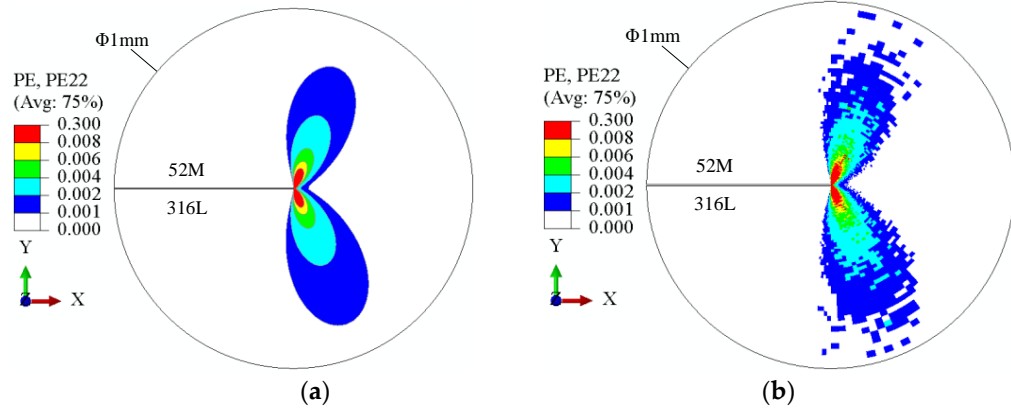

**Figure 15.** Normal plastic strain at the tip of crack 2: (**a**) homogeneous; (**b**) inhomogeneous.

Figure 16 shows the distribution of normal plastic strains for multiple observed paths in the homogeneous and inhomogeneous fusion boundary line specimens. The normal plastic strain from path 2 to the crack tip of path 5 can be seen in Figure 16a. The plastic strain curves for the inhomogeneous fusion boundary line specimens fluctuate more, and the range of curve fluctuations is smaller as a result of the lower degree of non-homogeneity on the Alloy 52M side, as can be seen when comparing the curves for path 2 in the homogeneous and inhomogeneous specimens in the figure. The fluctuating gradient of plasticity will become smaller as the radius of the circumference increases.

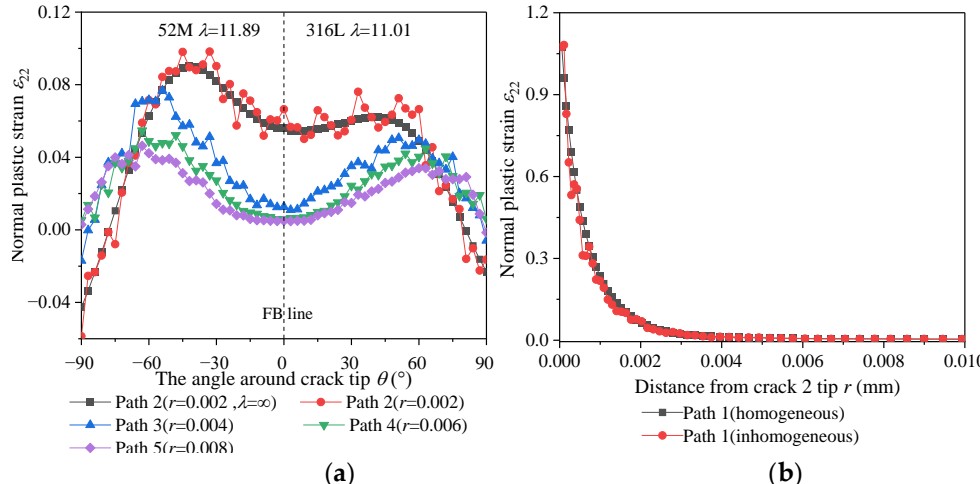

**Figure 16.** Normal plastic strain under homogeneous and inhomogeneous conditions: (**a**) around the crack 2 tip; (**b**) in front of the crack 2 tip.

Figure 16b shows the normal plastic strain along path 1 for the fused boundary line specimen. The plasticity in a similar inhomogeneous material will be greater than that in a homogeneous material. The strains in the Alloy 52M region are bigger than those in the 316L stainless steel region, and they are asymmetrically distributed on both sides of the fusion line crack tip in the homogeneous and inhomogeneous specimens. According to the results, the fusion line region is more susceptible to SCC phenomena than Alloy 52M and 316L stainless steel, and the cracks extend more quickly in the inhomogeneous material.

*3.5. Comparison of SCC Crack Growth Rate*

The Ford–Andresen model is based on the slip-dissolution/oxidation mechanism, and the SCC growth rate can be expressed as [14]:

$$\frac{da}{dt} = \kappa'_a \cdot \left( \frac{d\varepsilon_p}{da} \right)^{1/(1-m)} \tag{6}$$

where $\kappa'_a$ is the oxidation rate constant at the crack tip, determined by the material and electrochemical environment near the crack tip, $d\varepsilon_p/da$ is the change in the normal plastic strain at the characteristic distance $r_0$ in front of the crack tip, and $m$ is the exponent of the current density decay curve. $\kappa'_a$ is set to $7.478 \times 10^{-7}$, and m to 0.4 [25].

Currently, the SCC crack extension rate on DMWJ can be obtained by accurately calculating the normal plastic strain rate $d\varepsilon_p/da$ at the characteristic distance $r_0$. Among them, the choice of feature distance is crucial. Therefore, the appropriate choice of the characteristic distance $r_0$ is directly connected to the accuracy of the prediction results [15].

The SCC growth rates of homogeneous and inhomogeneous models with experimental data are displayed in Figure 17. The CGR in the homogeneous model lowers with increasing crack tip distance $r$ in a monotonically decreasing trend, while the growth rate changes from $1.2 \times 10^{-6}$ to $1.5 \times 10^{-7}$ mm/s, when $r_0$ changes from 4 to 11 μm. In contrast, the CGR in the inhomogeneous model tends to fluctuate up and down along the homogeneous model, with the growth rate fluctuating from $2.7 \times 10^{-6}$ to $2.0 \times 10^{-7}$ mm/s as $r_0$ changes from 4 to 11 μm. It is not difficult to find that the inhomogeneity of the material leads to the divergence of the predicted CGR results. According to the experimental data of Ruolin et al. [34] the homogeneous model is closer to the experimental data at $r_0 = 9.93$ μm for the same constant load. At $r_0 = 6.71$ μm, the inhomogeneous model more closely matches the experimental findings, and the difference between the two is nearly 1.5 times. This shows that the non-homogeneity of the material will affect the accuracy of crack extension rate prediction, and 6.71 μm is a more reliable choice among the inhomogeneous models.

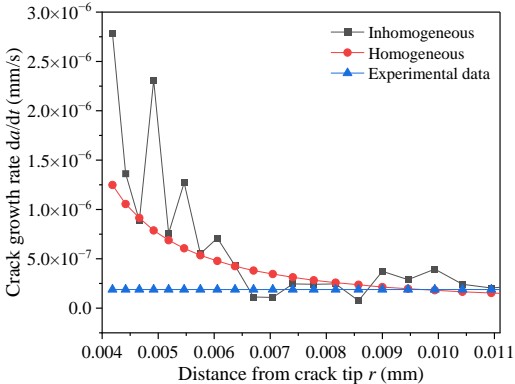

**Figure 17.** Comparison of SCC growth rate with experimental data.

## 4. Conclusions

In this study, the mechanical property parameters of Alloy 52M/316L stainless steel dissimilar metal welded joints (DMWJ) were obtained by physical experiments and combined with numerical simulation to obtain an inhomogeneous DMWJ model. The stress

field state and crack growth rate (CGR) of the stress corrosion cracking (SCC) tip at different locations in homogeneous and inhomogeneous specimens were compared and analyzed to further improve the method of determining the characteristic distance $r_0$. The main results are summarized as follows:

(1) Under the same constant load conditions, the inhomogeneity of the material's mechanical properties will lead to an asymmetric distribution of stress–strain at the crack tip at different locations. The higher the degree of non-uniformity, the greater the gradient of plastic strain at the crack tip.

(2) Combining the plastic strain zone at different locations of the crack in the homogeneous and inhomogeneous specimens, the crack has a higher plastic strain in the inhomogeneous specimen at the same constant load.

(3) When the crack is located at the interface between Alloy 52M and 316L stainless steel, it will provide a larger plastic strain for Type I (opening mode) cracks and is more likely to extend into the Alloy 52M region.

**Author Contributions:** Conceptualization, K.Z.; methodology, K.Z.; software, Z.W.; validation, B.W. and Z.W.; investigation, B.W. and Z.W.; data curation, B.W.; writing—original draft preparation, K.Z.; writing—review and editing, K.Z.; visualization, H.X.; supervision, H.X.; funding acquisition, K.Z. All authors have read and agreed to the published version of the manuscript.

**Funding:** This research was funded by the National Natural Science Foundation of China (Grant No. 52075434) and the Natural Science Basic Research Plan in Shaanxi Province of China (Grant No. 2021JM389).

**Data Availability Statement:** The data presented in this study are available on request from the corresponding author.

**Conflicts of Interest:** The authors declare no conflict of interest.

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
