# Peer review of "Effect of Material Inhomogeneity on the Crack Tip Mechanical Field and SCC Growth Rate of 52M/316L Dissimilar Metal Welded Joints"

_metals, doi:10.3390/met12101683_

Round 1

Reviewer 1 Report

According to the title, the paper should study the phenomenon of stress corrosion, especially the rate of crack growth. However, from the text of the article, it is obvious to the reviewer that the authors do not have sufficient knowledge of SCC to study this phenomenon let alone write scientific articles about it. The authors model SCC as a purely mechanical process without considering the influence of the corrosive environment, which must always be included. This article does not meet the scientific level and quality of the journal.

Specific comments:

1 Introduction: The level of risk of crack formation by SCC process depends not only on the mechanical properties of the material, but also on the quality of the water environment, especially the oxygen content of the water. You cannot write about susceptibility to SCC without mentioning the environmental conditions. The text needs to be completely rewritten.

Figure 6. The finite element model of the CT specimen is based on the erroneous assumption that the SCC crack front is rounded, as shown in (c) Details around the crack tip of the heterogeneous material. It is well known that the SCC crack tip is very sharp.

Figure 17: The model results are not in agreement with the experimental data. It is not stated from which source these data are from. It is not discussed why they are not in agreement with the model.

Conclusions: the text do not correspond to the results.

References – the reviewer do not believe that the authors have red all the papers

Reviewer 2 Report

The paper is adequate to publication with respect to the information of the SCC modeling and the materials used in pressurized water reactors in the nuclear power plants; but in order to get a suitable paper, I would suggest that you have someone that is very fluent in English work with you to correct the paper, and mainly you must do the follow corrections:

Page 1, line 41.  You can write in one brackets the references, for example [10-12].

Page 3, line 97.  You must explain what meaning the equation variables, like G.

Page 3, line 84, 102 and 106.  You must correct the references, use brackets. After this page, all references need to be corrected

Page 4, line 120. To correct the word Alloy in figure 2.

Page 4, line 128. To correct Table 21617, it can be Table 2.

Page 4, line 140. To correct Table head “material1716”.

Reviewer 3 Report

The present manuscript by Kuan Zhao, Bangwen Wang, He Xue, and Zheng Wang, entitled "Effect of material inhomogeneity on the crack tip mechanical field and SCC growth rate of 52M/316L dissimilar metal welded joints", reports the results of computer modelling of strain state and crack development in 52M nickel alloy - AISI 316L stainless steel dissimilar welds, as well as in the heat-affected zones of these materials. By the reviewer's estimation, the presented research is relatively novel and has a fair practical applicability. The manuscript itself leaves a very good impression. The title entirely corresponds with its contents. The abstract summarizes the contents of the manuscript in a comprehensive, but concise manner. The chapter 'Introduction' provides a deep enough overview of the research field and presents the motivation for the research clearly and convicingly. The chapter 'Methods and Calculation Model' generally gives a comprehensive description of the experimental procedure (including finite element modelling state), although some minor supplementations could still be made (these are specified below). The results, presented in the chapter 'Results and Discussion', are logically and thoroughly analyzed. The conclusions generally outline the findings well. At the same time, the reviewer would recommend to delete the conclusion no. 3 as non-informative, and split the conclusion no. 2 into two separate conclusions (a new conclusion could start from "Both homogeneous and inhomogeneous exhibit ..."). References are relevant and sufficient in number. It is also worth noting that 31.5% (eleven references out of 35) are from 2019 or later years, what is an indirect indication of a high relevance of this study. English is nearly errorless. Formatting is generally fine, too, although some small corrections would be recommended.

To conclude, the reviewer does recommend this manuscript for publication after a minor revision. The reviewer's comments, aimed at polishing the manuscript, are given below.

1. Line 15: it should be specified here, what 'Type I cracks' mean.

2. Line 34: 'organizational and mechanical properties inhomogeneity', not 'mechanical property inhomogeneity'.

3. Line 62: 'Ming et al. [22]', not 'Ming et al.[22]'.

4. Line 63: 'the mechanical properties of welded joints', not 'the mechanical property parameters of welded joints'.

5. Line 64: 'in the hardness of welded joints in the SA508-52M interface region' should be deleted for the sake of conciseness.

6. Line 84: '[25]', not '25'.

7. Line 86: 'material [23][24]' or 'material [23,24]', not 'material2324'.

8. Line 90: probably lambda should be used instead of m in Equation (1).

9. Line 98: 'the lower is the non-uniformity', not 'the lower the non-uniformity'.

10. Line 106: ASTM standard number must be specified here; also '[27]', not '27'.

11. Figure 1: thickness of the specimen should be specified.

12. Line 121: 'Welded', not 'welded'.

13. Line 124: '[20]', not '20'.

14. Line 128: 'Table 2 [16][17]' or 'Table 2 [16,17]', not 'Table 21617'.

15. Line 131: '[28]', not '28'; load, used for Vickers microhardness measurements, must be specified here.

16. Line 135: ISO standard number must be specified here; also '[29]', not '29'.

17. Line 137: '1 mm', not '1mm'.

18. Line 138: '2 mm', not '2mm'.

19. Line 140: 'Mechanical properties of the materials [16][17].' or 'Mechanical properties of the materials [16,17].', not 'Mechanical property parameters of the material1716'.

20. Table 2, 2nd column from the left: 'E (MPa)', not 'E(MPa)'.

21. Figure 4, y-axis: 'Vickers Hardness / HV0.1', not 'Vickers Hardness/HV0.1'.

22. Line 145: 'Peng et al. [30]', not 'Peng et al.30'.

23. Line 149: '[31]', not '31'.

24. Line 152: 'yield strength', not 'yield Strength'.

25. Line 172: apparently 'fit', not 'Fit'.

26. Line 175: apparently 'write', not 'Write'.

27. Line 189: 'm^(1/2) [32]', not 'm^(1/2)32'.

28. Line 191: '[33]', not '33'.

29. Line 193: '10 [mue]m [34]', not '10[mue]m 34'.

30. Line 194: '1 [mue]m to 10 [mue]m', not '1[mue]m to 10[mue]m'.

31. Line 195: '8 [mue]m', not '8[mue]m'.

32. Line 194: 'path, (c)', not 'path(c)'.

33. Line 214: 'MPa m^(1/2)', not 'MPa m1/2'.

34. Line 226: '0.2%: (1) K1=10, (2) K2=20, (3)K1=30, (4) K1=40 MPa m^(1/2) (a) Homogeneous', not '0.2%:(1) K1=10,(2) K1=20,(3) K1=30,(4) K1=40 MPa m^(1/2))(a) Homogeneous'.

35. Line 237: apparently 'Figure 8', not 'Fig. 8'.

36. Line 244: '8 [mue]m', not '8[mue]m'.

37. Line 245: '2 [mue]m', not '2[mue]m'.

38. Figure 10, x-axis: 'tip [theta] ([deg])', not 'tip [theta]([deg])'.

39. Line 268: 'Alloy 52M, the', not 'Alloy 52M. The'.

40. Lines 289-290: 'the higher is the degree of inhomogeneity, the greater is the fluctuation of the normal plastic strain, the greater is the impact on the accuracy of predicting the crack extension rate', not 'the higher the degree of inhomogeneity, the greater the fluctuation of the normal plastic strain, the greater the impact on the accuracy of predicting the crack extension rate'.

41. Figure 13, x-axis: 'tip [theta] ([deg]]', not 'tip [theta]([deg])'.

42. Line 310 and Figure 14: the strain extension actually seems to be higher in the AISI 316L steel zone, not in the 52M alloy.

43. Line 346: 'as [14]' , not 'as14'.

44. Line 351: 'k'a is set to 7.478x10^(-7)', not 'Therefore set k'a to 7.478x10^(-7)'.

45. Line 356: 'results [15]', not 'results15'.

46. Line 358: apparently 'Figure 17', not 'Fig. 17'.

47. Line 360: '1.5x10^(-7) mm/s, when', not '1.5x10^(-7) when'; also '11 [mue]m', not '11[mue]m'.

48. Line 363: '11 [mue]m', not '11[mue]m'.

49. Line 365: 'al. [35]', not 'al.35'.

50. Line 366: apparently 'r0=6.71 [mue]m', not 'r0=6.71 m'.

51. Line 396: '6.71 [mue]m', not '6.71[mue]m'.

52. Lines 375, 376, 377: the abbreviations 'DMWJ', 'CGR', and 'SCC' could be explained here again (for those readers, who read only conclusions of an article).

53. Lines 382-383: 'the higher is the degree of non-uniformity, the greater is the gradient of plastic strain', not 'the higher the degree of non-uniformity, the greater the gradient of plastic strain'.

54. Lines 386-387: what was meant under 'Both homogeneous and inhomogeneous exhibit'? The reviewer requests to reformulate this wording.

55. Line 388: the term 'Type I crack' should be explained here in more detail.

56. The formatting style of the references must be checked, and the formatting of the references must be uniform. More specifically, the reviewer requests to check

a) whether comma or semicolon should be used for separation of authors' surnames;

b) whether journal title must be written in full or abbreviated;

c) whether hyphen or en dash (the authors may refer to [1] or another relevant source for more information) should be used for designation of page ranges.

57. Lines 415-416: 'high-temperature water', not 'hightem-perature water'.

58. Reference no. 14 (line 424): the page range or article number must be added.

59. References no. 15 (line 426), no. 34 (line 467): [J] should apparently be inserted instantly after the journal title.

60. Line 430: '[J]', not '[C]'; also 'Advanced Materials Research, 2012,' or 'Adv. Mater. Res., 2012,', not 'Advanced Materials Research. Trans Tech Publications, 2012,'.

61. Reference no. 25 (lines 448-449): inverted commas around the article title must be deleted.

62. Line 457: 'ISO 6507-1, Metallic', not 'ISO 6507-1,Metallic'; also a full stop must be inserted at the end of the reference.

63. Line 463: apparently 'The Pressure Vessels', not 'the Pressure Vessels'.

64. Reference no. 33 (lines 464-465): volume number must be inserted.

65. Line 467: 'Vessels', not 'V essels'.

[1] https://grammarist.com/usage/hyphen-en-dash-or-em-dash (accessed 2022-9-8).1
